# Internal and forced ocean variability in the Mediterranean Sea

Roberta Benincasa[1], Giovanni Liguori[2,3], Nadia Pinardi[1,3], and Hans von Storch[4]

[1]University of Bologna, Department of Physics and Astronomy, Bologna, Italy
[2]University of Bologna, Department of Biological, Geological, and Environmental Sciences, Bologna, Italy
[3]Centro EuroMediterraneo sui Cambiamenti Climatici, Lecce, Italy
[4]Helmholtz-Zentrum Hereon, Institute for Coastal Systems, Geesthacht, Germany

**Correspondence:** Roberta Benincasa (roberta.benincasa2@unibo.it)

**Abstract.** Two types of variability are discernible in the ocean: a response to the atmospheric forcing and the so-called internal/intrinsic ocean variability, which is associated with internal instabilities, nonlinearities and the interactions between processes at different scales. Producing an ensemble of 20 multi-year ocean simulations of the Mediterranean Sea, initialized with different realistic initial conditions, but using the same atmospheric forcing, the study examines the intrinsic variability in terms of its spatial distribution and seasonality. In general, the importance of the external forcing decreases with depth but dominates in extended shelves such as the Adriatic Sea and the Gulf of Gabes. In the case of temperature, the atmospheric forcing plays a major role in the uppermost 50 m of the water column during summer and the uppermost 100 m during winter. Additionally, intrinsic variability displays a distinct seasonal cycle in the surface layers, with a prominent maximum at around 30 m depth during the summer connected to the summer thermocline formation processes. Concerning current velocity, the internal variability has a significant influence at all depths.

## 1 Introduction

High-resolution numerical ocean models are currently employed in operational ocean forecasting, providing short-term forecasts with a lead time of 14 days for both global and regional seas, as described by Le Traon et al. (2019) and Coppini et al. (2023). Consequently, comprehending the uncertainties associated with these forecasts is crucial for enhancing future ocean forecasting. Model uncertainties restrict the duration for which accurate predictions can be made and hinder the practical applications that rely on these forecast products.

Lorenz (1975) identified two types of uncertainties affecting predictability in the climate system: the predictability of the first kind derives from the resolution of an initial value problem, i.e. predicting future states of the system given the same external forcing and different initial conditions, whereas the predictability of the second kind is connected to the boundary value problem and deals with the response of the system to changes in the external forcings. In the uncoupled ocean prediction problem, the predictability of the second kind is connected to the atmospheric fluxes at the air-sea interface while the predictability of the first kind is connected to the mesoscale and submesoscale flow field that in turn is generated by mean flow instabilities and eddy-mean flow interaction processes (Soldatenko and Yusupov (2017)).

The understanding of how internal variability can affect climate predictability, and justify the observed red profile of the climate

variance spectra, was first achieved by Hasselmann (1976). To demonstrate the importance of internal variability in climate models, Hasselmann formulated a stochastic climate model whose main assumption is that the climate system may be divided into rapidly varying, random components and a slowly responding part. Climate variability is then shown to be due to the internal random components. The slow component behaves as an integrator of these inputs, whereas the fast component supplies the slow component with energy allowing the existence of internal variability in the climate system. Moreover, Hasselmann proved that climate variability would grow indefinitely without a stabilizing internal feedback mechanism. Consequently, the investigation of climate variability must be shifted from looking for positive to negative feedbacks that allow the climate system to reach stationarity in the absence of any external forcing. In the same years as Hasselmann's study, mesoscale eddies and flow instabilities were mapped for the first time in the ocean (Harrison and Robinson (1978), McWilliams (1996)) and the presence of intense ocean internal variability was verified to exist.

The *internal variability*, or equivalently *intrinsic* or *stochastic* variability, is ubiquitous in the climate system and it is due to both the nonlinearity and the numerous degrees of freedom of climate itself (von Storch et al. (2001)). Understanding the internal variability enables us to statistically determine if a change is consistent with internal variability or instead at least partly related to external factors. Investigations on internal ocean variability with an ensemble approach were first found in literature starting in the 2000s when eddy-resolving global ocean models were introduced (Jochum and Murtugudde (2004), Jochum and Murtugudde (2005), Arbic et al. (2014), Sérazin et al. (2015), Bessières et al. (2017), Leroux et al. (2018)). The first extensive study on the role of internal variability in the global ocean was done by Penduff et al. (2018). They showed that mesoscales compose the ocean stochastic elements of the climate system and highlighted the necessity to adopt ensemble methods. Moreover, they showed that in several areas the internal ocean variability is more significant than atmospheric variability as a contribution to climate variability for both "the low-frequency variability and the long-term trends of regional ocean heat content". Similarly, in the study by Hogan and Sriver (2019) it was shown that internal variability is fundamental in setting the time scale for the ocean temperature adjustment process, increasing the speed at which the ocean takes up heat from the atmosphere, that is instead highly underestimated by just considering atmospheric variability. Lastly, as regards the scale dependency of the internal variability, it was demonstrated by Tang et al. (2019) that additional intrinsic variability is produced by increasing the horizontal spatial resolution of ocean models from 1° to 0.04°. Furthermore, Tang et al. (2020) analyzed the ratio of the externally forced response and the internally generated variability in the South China Sea and showed that the external forcing is dominant at large scales, while most of the variability is internally generated at smaller scales.

Our paper addresses the predictability of the first kind for the Mediterranean Sea, focusing on the internal variability, following the methodology of Penduff et al. (2018), Leroux et al. (2018), Tang et al. (2020) and Lin et al. (2022).

The Mediterranean Sea is a semi-enclosed basin with an average depth of 1500 m and it is connected to the Atlantic Ocean through the Strait of Gibraltar and to the Marmara Sea through the Dardanelles. The external forcings are the wind stress, responsible for the permanent gyres of the basin (Pinardi et al. (2015)), and the heat and water fluxes that control the overturning circulation (Pinardi et al. (2019)). Furthermore, ocean circulation in the Mediterranean Sea is characterized by significant interactions between different scales (Robinson et al. (2001)) and mesoscale variability is intense, producing 50 - 60 % of ocean kinetic energy variability (Bonaduce et al. (2021)). Mesoscales, for example, are associated with instabilities of the main flow

and they have a role in water mass transport across the basin (Demirov and Pinardi (2007)) and deep water formation processes enhancing its intrinsic variability (Waldman et al. (2017a), Waldman et al. (2017b)). In addition, Waldman et al. (2018) studied the intrinsic variability of deep water formation processes in the North Western Mediterranean Sea finding that it contributes significantly to DWF interannual variability and is mostly generated by baroclinic instability.

We use a well-calibrated version of the Mediterranean Sea general circulation model used for short-term forecasting and we produce an ensemble of simulations using the same atmospheric forcing but different realistic initial conditions. First, we characterize the 3-D spatial distribution and the seasonal characteristics of the internal ocean variability over the entire basin. Secondly, we quantify the relative importance of the internal variability with respect to the atmospherically forced response using the *noise-to-signal ratio*. In the current analysis, the *signal* is identified with the mean of the ensemble of simulations whereas the *noise* or internal variability is approximated by the standard deviation of the ensemble members with respect to the mean. The questions we want to answer are: how large is the internal variability and what is its structure in the Mediterranean basin? What is the ratio between noise and signal?

The rest of the manuscript is organized as follows. The experimental setup is described in Section 2, alongside with the explanation of the rationale for the simulations and the statistical methods used. In Section 3 we show the evaluation of the quality of the ensemble experiment and in Section 4 the internal variability's spatial distribution, seasonality and dependence on depth are shown. Lastly, in Section 5 the stochastic variability is compared to the atmospherically forced variability and their relative importance is assessed. Conclusions are summarized in the last section.

## 2    Model set up and simulations

The model used in this work is one of the operational versions of the forecasting system of the Mediterranean Sea (Clementi et al. (2019), Coppini et al. (2023)) consisting of a coupled general circulation-wave model (Fig. 1) without tidal components. For the present work the wave component was disregarded. The model horizontal grid resolution is 1/24°(ca. 4 km) and has 141 unevenly spaced z* levels. The model is forced by momentum, water and heat fluxes computed through bulk formulae using the operational analysis and forecast fields from the European Centre for Medium-Range Weather Forecasts (ECMWF). The ECMWF atmospheric boundary conditions have a horizontal resolution of 1/8° up to December 2020 and of 1/10° after. This change in the forcing's horizontal resolution is irrelevant since during the analyzed period all simulations are forced by the same atmospheric fields. For more details on the specific model implementation, refer to Coppini et al. (2023).

 Following the idea in Penduff et al. (2018) and in Tang et al. (2020), an ensemble of 20 simulations of the Mediterranean Sea is generated with the same atmospheric conditions, but with different start dates and consequently different run times. Each simulation is initialized every three months starting from January 2016 to October 2020. The initial conditions are taken from the Copernicus Marine Service analyses (Clementi et al. (2019)) and all simulations last up to December 2021, as explained in Figure 2. Thus, the ensemble spread, related to internal variability, is generated by the different initial conditions. It is clear that the further back is the start date of the simulation, the longer has passed from the last analysis and internal non-linearities have started to deviate the solution from it. The same results could be obtained by just adopting different computing platforms,

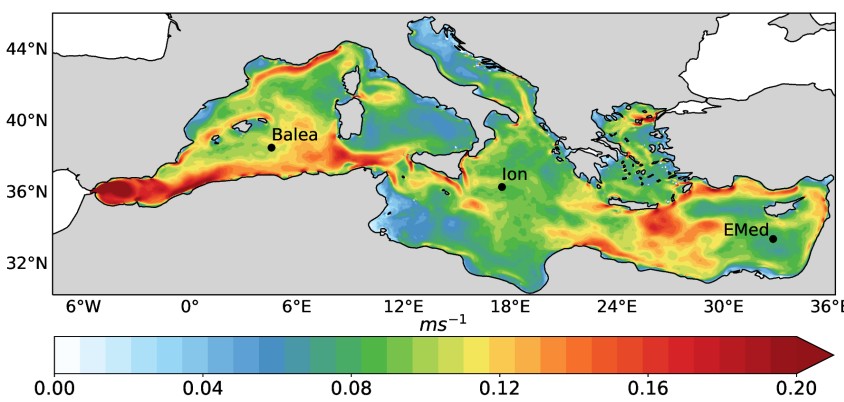

Average $\sigma_R$ for current speed from 1987-2021 reanalyses at the surface

**Figure 1.** Average interannual spread $\sigma_R$ in the Mediterranean Sea at the surface computed with the reanalyses from the period 1987 - 2021 (see Section 3). $\sigma_R$ is computed as the spread of an ensemble composed of the current speed field of each year from 1987 to 2021. The dots indicate the locations that were chosen for the analysis: Balearic Islands (**Balea**), Ionian Sea (**Ion**) and Eastern Mediterranean Sea (**EMed**).

as proved in Lin et al. (2023a), since what is needed is just small disturbances. It is important to notice that the choice of having an ensemble of 20 members was somewhat arbitrary, even if it was the largest number of members compatible with our

computational resources and returned results similar to a smaller ensemble of only 5 members (Figure S4 and Figure S5 in the Supplementary Material).

In addition, each ensemble member is driven through bulk formulae by identical atmospheric forcing function and atmospheric conditions, but variations in air-sea fluxes within the ensemble arise due to differences in oceanic variables (Bessières et al. (2017)). This configuration gives realistic results, but induces, for instance, an implicit relaxation of the SST towards the same

equivalent air temperature, thereby damping the SST spread (Barnier et al. (1995)).

The analysis is performed over the entire Mediterranean Sea at fixed depth levels (1 m, 20 m, 50 m, 100 m, 200 m, 500

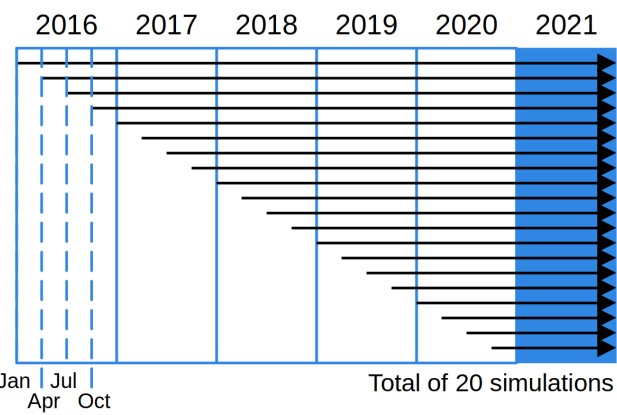

**Figure 2.** Scheme of the ensemble of 20 multi-year ocean simulations (black arrows). The blue box indicates the analyzed year.

m, 1000 m) and at some locations, indicated by the dots in Fig. 1, that are chosen to characterize the entire basin from the Western to the Eastern side. Limiting the analysis to the first 1000 m of the water column was justified in order to capture a meaningful *N/S*. The deeper levels showed decreased spread and decreasing atmospheric influence, as expected for the large-scale circulation. Despite its arbitrariness, this boundary ensures a practical balance between vertical variability and statistical significance. Furthermore, starting from daily outputs, we focus on the seasonal timescale, in particular considering the four-month winter and summer seasons defined in Artegiani et al. (1997) for the Mediterranean region, i.e. January-April (**JFMA**) and July–October (**JASO**), respectively.

Lastly, it is essential to acknowledge that the model we used is not everywhere eddy-resolving but mainly eddy-permitting. This is a consequence of the fact that the first Rossby radius of deformation in the Mediterranean Sea varies from 3 to 13 km (Beuvier et al., 2012) with larger values in the basin's interior and the southern areas. In contrast, in the Adriatic Sea and the Gulf of Gabes, the Rossby radius is generally smaller than the model's horizontal resolution thus possibly artificially decreasing the importance of internal variability.

## 2.1 Statistical methods

We employ basic ensemble statistics to measure the internal and forced variability of the ocean. The ensemble mean of the simulations is considered to represent the forced response of all members to the common forcing, hereafter indicated with the signal. Subtracting this quantity from each member we obtain the stochastic variations, uncorrelated between the members (Penduff et al. (2018), Leroux et al. (2018)). Consequently, the internal ocean variability $\sigma_I$ can be computed considering the discrepancy of each member from the ensemble average, i.e. the *ensemble standard deviation* or *ensemble spread* or *noise*, at each grid point $(i,j)$:

$$\sigma_I(i,j,t) = \sqrt{\frac{1}{N-1}\sum_{n=1}^{N}[f_n(i,j,t) - f_{mean}(i,j,t)]^2} \tag{1}$$

where $N = 20$ is the ensemble size, $f_n$ is the n-th member and $f_{mean}$ is the ensemble mean. On the other hand, the atmospherically forced variability $\sigma_A$, that is the variability shared by all simulations, is given by the evolution of the single-member means expressed by its temporal standard deviation:

$$\sigma_A(i,j;\tau) = \sqrt{\frac{1}{\tau-1}\sum_{t=1}^{\tau}[f_{mean}(i,j,t) - \bar{f}_{mean}(i,j)]^2} \tag{2}$$

where $\bar{f}_{mean}$ is the temporal average of the ensemble mean over the chosen period $\tau$, i.e. 120 days corresponding to a season. In the present work, we use internal or intrinsic variability and ensemble spread interchangeably, the latter being the mathematical formulation of the former. To compare (1) and (2), we consider the ratio, called *noise-to-signal ratio*, as the temporal average of the internal variability over the forced variability during the same period:

$$\frac{N}{S}(i,j;\tau) = \frac{\langle\sigma_I(i,j,t)\rangle_\tau}{\sigma_A(i,j;\tau)} \tag{3}$$

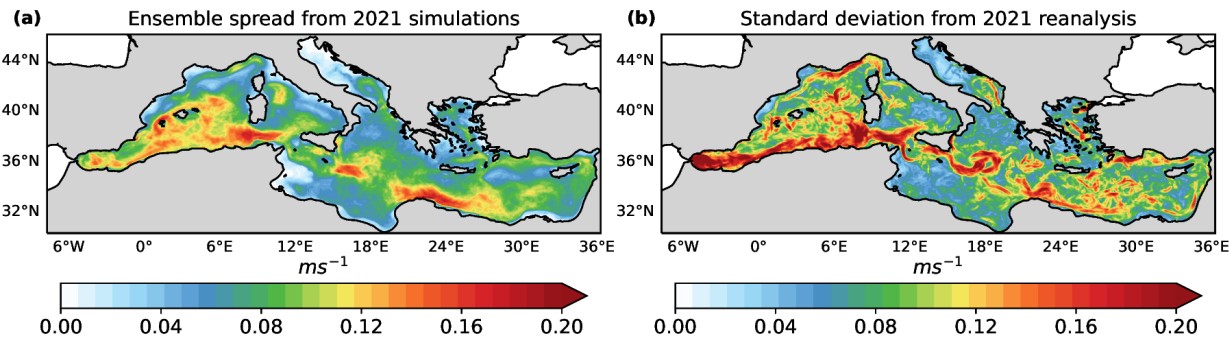

**Figure 3.** Comparison between the winter averaged ensemble spread $\sigma_I$ with $N = 20$ **(a)** relative to current speed computed from the ensemble of simulations and the corresponding seasonal standard deviation $\sigma_{2021}$ **(b)** obtained from the 2021 reanalysis.

## 3   Variability: model output compared to reanalysis

To evaluate the quality of the model simulations, the ensemble members are compared to the reanalysis of the year 2021 (Escudier et al. (2021)) produced approximately with the same model but assimilating drifting profiles, satellite sea surface temperature, and altimetry.

For current velocities, we compared the internal variability found in the simulations and the standard deviation computed from the 2021 reanalysis, hereafter called $\sigma_{2021}$. We would like to verify that $\sigma_I$ is not overestimated when compared to the natural variability represented by the standard deviation of the reanalysis. We found that the ensemble members' internal variability is generally smaller than both the interannual variability $\sigma_R$ computed from the 1987 - 2021 reanalyses (Fig. 1) and the standard deviation $\sigma_{2021}$ of the 2021 reanalysis (Fig. 3), with the only exception of the region around the Balearic Islands. We argue that the ensemble internal variability is comparable with the natural variability of the Mediterranean Sea at all depths and seasons.

Another evaluation of the ensemble spread is done by comparing it to the RMSE of the ensemble mean with respect to the 2021 reanalysis $r$, as suggested in Fortin et al. (2014):

$$\sqrt{\overline{\sigma^2}} = \sqrt{\frac{N}{N+1}} RMSE \tag{4}$$

where $\sqrt{\overline{\sigma^2}} = \sqrt{\frac{1}{\tau}\sum_{t=1}^{\tau} var(t)}$ and $RMSE = \sqrt{\frac{1}{\tau}\sum_{t=1}^{\tau}(f_{mean}(t) - r(t))^2}$, with $\tau \to \infty$ since this relation holds for large values of $\tau$ and this is the reason we considered $\tau = 365$ days corresponding to the entire 2021. It is important to note that both Eq. 4 and 1 depend on the ensemble size $N$. Figure 4 (and Figure S1 in the Supplementary material) show the values of the mean ensemble variance versus the RMSE for temperature and current speed, respectively, for each grid point averaged over the entire year and at different depths. Points above the line described by Eq. 4 characterize an overdispersive ensemble, whereas those below show an underdispersive ensemble. Overall, the present ensemble can be considered underdispersive. For temperature, the ensemble variance and the uncertainty peak at roughly 30 m depth (Fig. 4), and in the case of current speed, both the ensemble variance and the RMSE decrease with depth (Fig. S1).

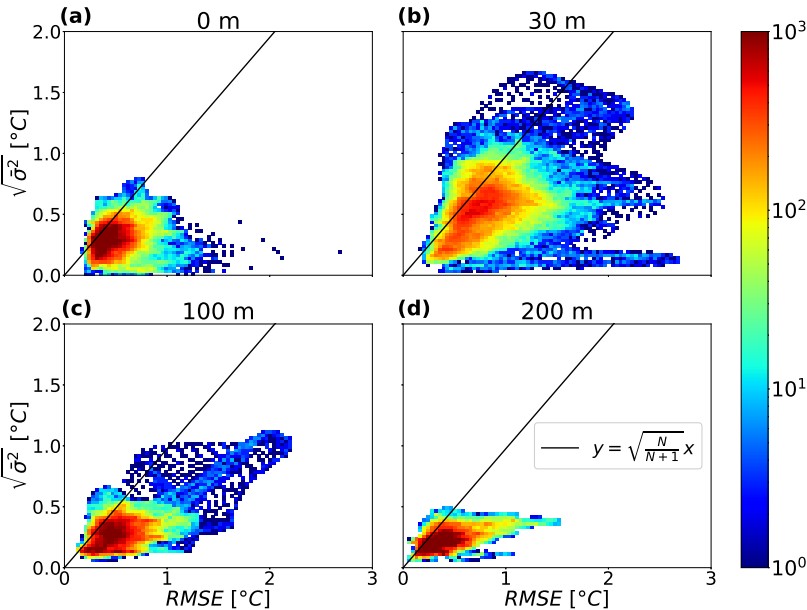

**Figure 4.** Ensemble dispersion relative to sea potential temperature: verification of Equation 4 averaged over the entire year 2021 for each grid point at the surface **(a)**, at 30 m **(b)**, at 100 m **(c)** and at 200 m **(d)**. The black line indicates the ideal relation expressed in Eq. 4.

Lastly, the ensemble spread and the noise-to-signal ratio were computed also for the year 2020 (see Fig. S16-S19 in Supplementary material), but with only 16 simulations, resulting in the same findings as for 2021, proving that these are not peculiar to the chosen year.

## 4 Characterization of internal variability

The ensemble spread for the current speed $v$ and potential temperature $T$ is shown in Fig. 5 at 30 m depth for both seasons (see Fig. S6-S7 in Supplementary material for all depth levels). In the literature, Hecht et al. (1988) have described the Eastern Levantine thermocline seasonal variations showing it to be located between 20 and 40 m depth. Thus, we refer to 30 m as the average depth of the seasonal thermocline.

It is evident that internal variability for currents is high amplitude at the surface and that in both seasons its spatial pattern is unchanged with depth while decreasing in intensity. In winter, the areas with the highest intrinsic variability (0.12 - 0.15 $ms^{-1}$) are distributed along the southern parts of the basin while in summer, an equally significant presence of internal variability is found in localized areas: in the Ionian Sea and in the westernmost part of the basin. In both seasons, the Adriatic Sea and the Gulf of Gabes show the minimum values of the ensemble spread (less than 0.03 $ms^{-1}$).

For temperature, the ensemble spread exhibits a much more pronounced seasonality. In winter, the maximum values are of the order of 0.5 °C and are mainly located in the Eastern Mediterranean. Moreover, the spatial distribution of the spread remains

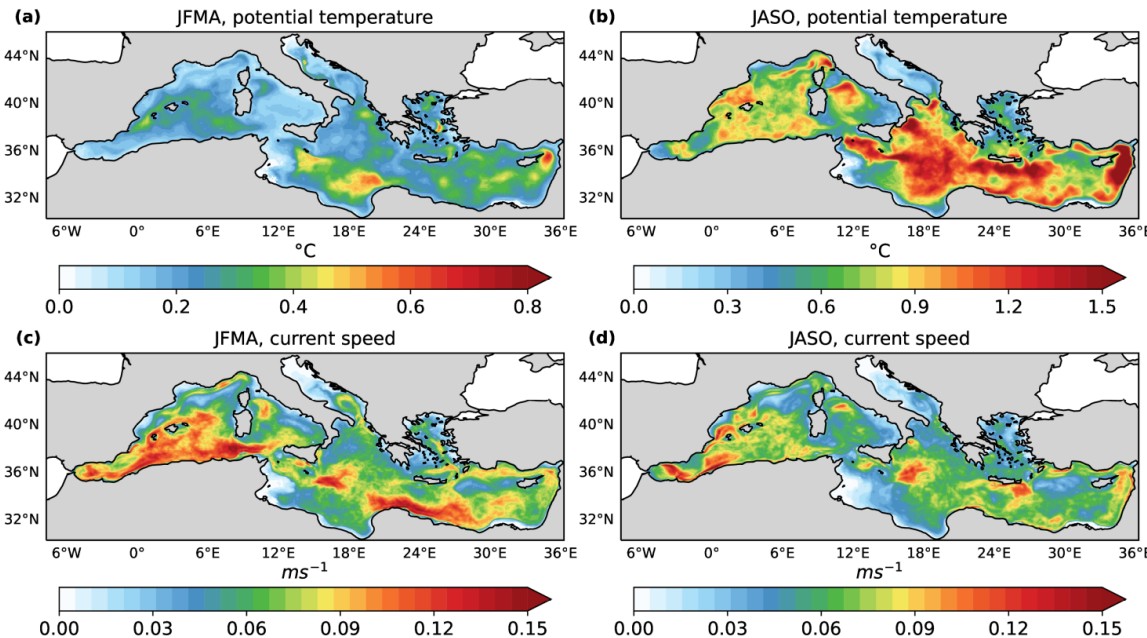

**Figure 5.** Seasonally averaged ensemble spread at 30 m depth for seawater potential temperature both in winter **(a)** and in summer **(b)**. Similarly for current speed in **(c)** and **(d)**.

practically constant along the water column up to the depth of 100 m. During summer the ensemble spread is largest at depths between 20 and 30 m (about 1.5 °C), which coincides with the upper seasonal thermocline center depth. The spread is again low in the Adriatic Sea and the Gulf of Gabes extended shelf areas. In fact, over shelf areas, the external forcing or signal exerts a more significant influence with respect to deep ocean areas (Tang et al. (2020), Lin et al. (2022)). For depths greater than 170   100-200 m, the seasonal trend disappears for both current and temperature spread (see Fig. S8 in Supplementary Material).

The vertical profile of the ensemble spread follows the vertical temperature gradient, as shown in Fig. 6. It peaks at the seasonal thermocline center depth, i.e. at the depth of the vertical temperature gradient maximum. In other words, the ensemble spread is greater where strong temperature gradients are present since small differences among the simulations are amplified by the significant changes in the temperature field at these depths. However, the relation between ensemble spread and vertical 175   temperature gradient varies depending on the location: in the Ionian Sea (Ion in Fig. 6) and in the Balearic Islands (Balea), the alignment between the two curves is notably robust. Conversely, in the Eastern Mediterranean (EMed), the peak of the temperature gradient occurs at approximately 20 m, whereas the maximum of $\sigma_I$ is found at 40 m. We argue that, during summer, thermocline processes exhibit significant internal variability and the spread observed at the peak of the vertical temperature gradient may arise from various mechanisms. First, baroclinic instability localized there can generate internal variability. Sec-180   ondly, changes in the position and strength of eddies can cause upwelling or downwelling, thereby influencing the mixed layer depth and consequently, the mid-depth of the thermocline (Figures S10 and S11 in the Supplementary material).

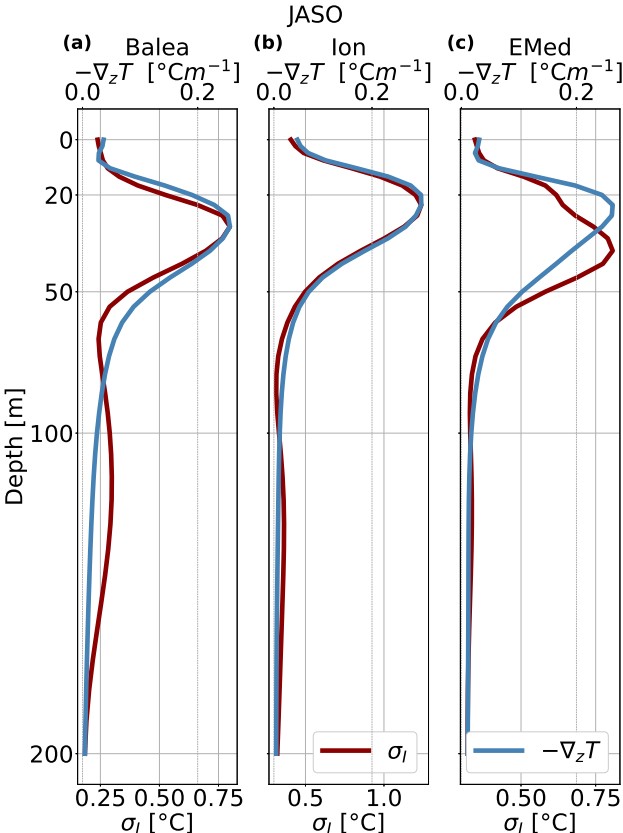

**Figure 6.** Seasonally averaged vertical profile of the ensemble spread $\sigma_I$ (red) for potential temperature and of the vertical temperature gradient $-\nabla T_z$ (blue) in summer at **Balea (a)**, **Ion (b)** and **EMed (c)**.

## 5 Comparison between noise and signal

In order to quantify the relative importance of the internal variability with respect to the atmospherically forced response, the noise-to-signal ratio (Eq. 3) is computed. Figure 7 summarizes the basin-averaged vertical profile of *N/S* across the two seasons
at the discrete depth levels defined in Section 2. The *N/S* for *T* is smaller than 1 up to 100 m (about 0.3 at the surface) and it increases with depth. In the surface layers, it shows greater values in winter, whereas at greater depths it attains systematically larger values (approximately equal to 6) in summer. On the other hand, the current speed's *N/S* is always greater than 1: it rises steadily in the first 50 m, reaching a maximum at 200 m in summer and winter, and then slightly decreases with depth. Thus, even though $\sigma_I$ has its maximum at some intermediate depth for *T* and it is maximum at the surface for *v*, the *N/S* is
greater at depth due to the diminishing importance of the atmospheric forcing with increasing depth. Moreover, whereas for *v* the internal variability is always dominant, for *T* the atmospheric response has a dominant influence in the surface layers, especially in winter.

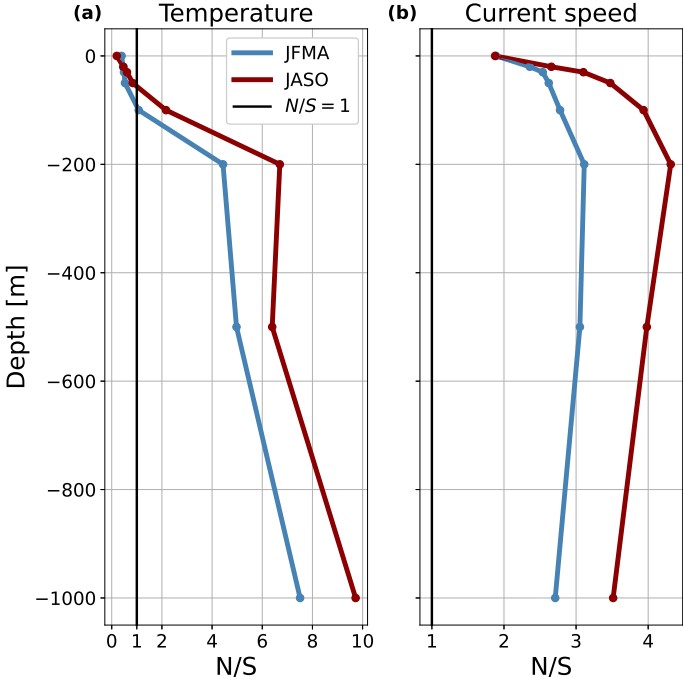

**Figure 7.** Vertical profile of the seasonally and spatially averaged noise-to-signal ratio for potential temperature **(a)** and current speed **(b)**. Red represents summer, whereas blue is winter and the black vertical lines correspond to $N/S = 1$.

As regards the spatial distribution of the noise-to-signal ratio, as expected the Adriatic Sea and the Gulf of Gabes are the
regions with the smallest values of *N/S* for both *T* and *v*. For potential temperature (see Fig. 8) *N/S* in summer is less than 1
over the entire Mediterranean Sea at the surface, but it increases with depth, especially along the African coast in the Western
Mediterranean and in the northern part of the Eastern Mediterranean. In winter, instead, in the Ionian Sea, north of the Gulf
of Sidra, *N/S* is greater than unity and it tends to grow with depth mainly in this region and along the coast of Spain in the
Western Mediterranean. Conversely, the model resolved mesoscale activity contributes to $\sigma_I$ for *v* in vast open ocean areas
of the basin. Figure 9, shows that *N/S* is higher than 1 in large open ocean areas offshore to vigorous northern and southern
boundary currents (northern Liguro-Provencal current, the Algerian Current, the Asia Minore current, etc.). We argue that this
internal variability is due to mesoscales. Some of the intense current regions, such as the Gibraltar inflow current and part of
the Liguro-Provencal current show a relevant contribution from external forcings, the Atlantic water inflow for the former and
the wind stress curl for the latter, as documented by many authors (Herbaut et al. (1997), Molcard et al. (2002)). In the shallow
areas of the Adriatic and Gulf of Gabes, where the external forcing is dominant, the mean flow probably fluctuates with the
forcings and prevents the onset of instabilities and the production of internal variability.

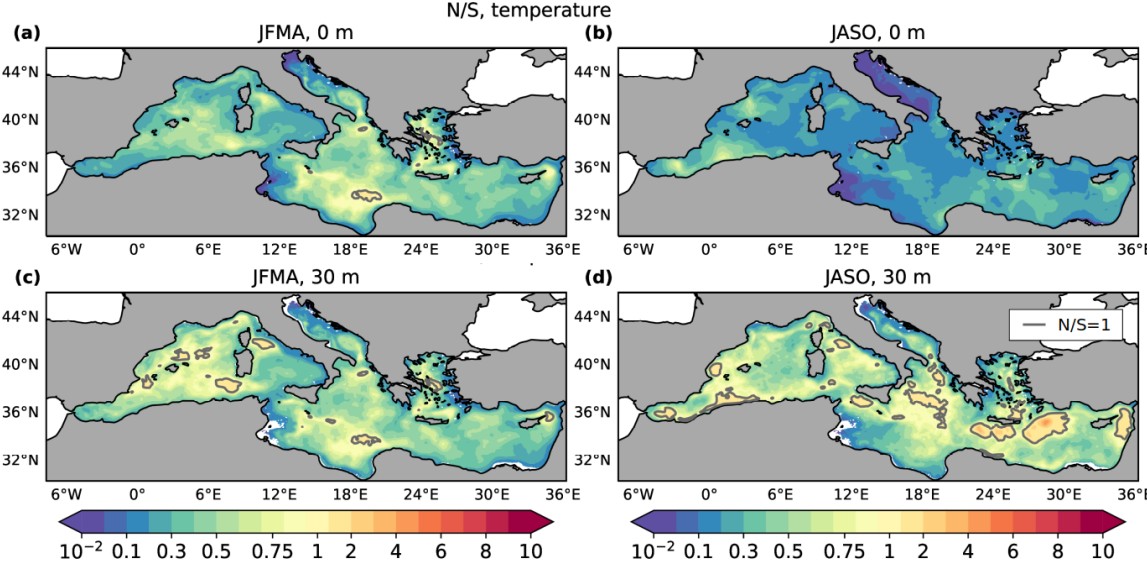

**Figure 8.** Seasonally averaged *N/S* for seawater potential temperature both in winter at the surface **(a)** and at 30 m depth **(c)**. Analogously for summer in **(b)** and **(d)**, respectively. The gray lines represent the isolines at *N/S* = 1.

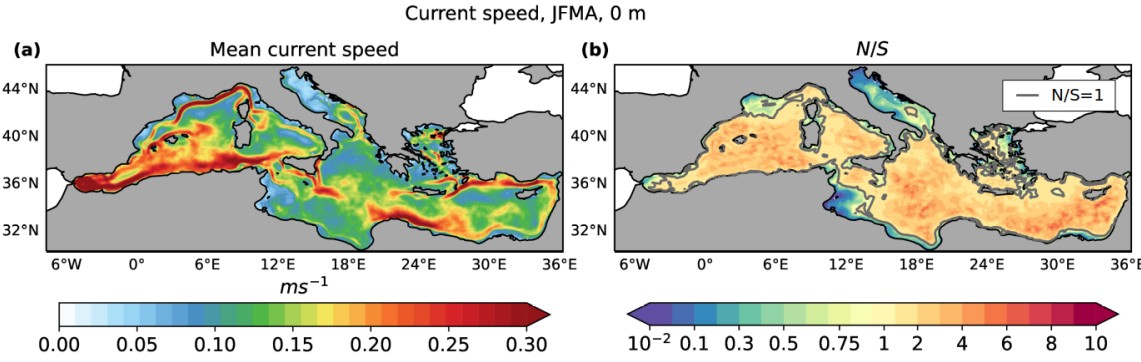

**Figure 9.** Seasonal average (winter, JFMA) at the surface of the ensemble mean **(a)** and of the *N/S* **(b)** for the speed of the current. The gray lines in the right plot represent the isolines at *N/S* = 1.

## 6   Discussion and conclusions

In the present work, we used a simulation ensemble approach to study, for the first time in the Mediterranean Sea, the internal variability or noise versus the atmospherically forced signal. Producing an ensemble of 20 simulations of the Mediterranean Sea in 2021 with different start dates but forced by the same atmospheric conditions, we were able to characterize the internal ocean variability $\sigma_I$ as opposed to the forced response of the ocean. Moreover, we used the noise-to-signal ratio *N/S* to measure the relative importance of the former compared to the latter. We characterized and quantified the internal variability and *N/S*

as regards their vertical profiles, seasonal cycle and spatial patterns for temperature and current speed. In general, the atmospherically forced response tends to decrease with depth and thrives instead in the extended shelf areas of the basin such as the Adriatic Sea and the Gulf of Gabes. Relative to temperature, the atmospheric forcing is dominant in the first 50 m of the water column in summer and in the first 100 m in winter. Furthermore, internal variability is dominant in most of the open ocean regions of the Mediterranean Sea, due to the intense mesoscale variability, offshore to intense northern and southern mean current systems. To note, some of the most intense currents have components that are externally forced, such as the Alboran current and segments of the Liguro-Provencal current system. Lastly, the intrinsic variability shows a large seasonal cycle in temperature with a sharp maximum in summer at roughly 30 m depth and a much less pronounced one at about 100 m depth in winter. The vertical profile of the spread in summer is probably related to the large variability in the upper thermocline due to internal variability. Regarding current speed, the internal variability is dominant at all depths and largest at the surface.

The assessment of the internal oceanic variability is crucial for tackling the issue of estimating the ocean's intrinsic predictability and comprehending the distinct roles of internal instabilities and external forcings in shaping ocean dynamics. Furthermore, this research serves as an additional validation of the efficient yet straightforward nature of this ensemble statistics, derived from the theory of stochastic climate models, whose aim is not to investigate specific hydrodynamic processes but to study the system's properties and statistics (Lin et al. (2023b)). Moreover, the noise-to-signal ratio proves to be an effective diagnostic indicator. Last but not least, the production of internal variability itself has been assessed, further proving the main idea behind stochastic climate models theory: in a dynamical system featuring the coexistence of diverse temporal scales and the possibility to distinguish between transient and mean components, variations can arise from their internal interactions without implying any external factor.

A limitation of our study is the underestimation of the internal variability stemming from the model's horizontal resolution of 1/24° which results in being too coarse for resolving mesoscale eddies everywhere in the Mediterranean Sea. Such underestimation could be particularly significant in the Adriatic Sea and the Gulf of Gabes where the spatial scales of mesoscale eddies tend to be smaller than the model's horizontal resolution. Thus, this could be an additional factor causing the small values of the ensemble spread in these regions. Nonetheless, the present study shows the importance of internal processes as opposed to the atmospheric influence compatible with the model resolution.

An extension of this analysis could involve the incorporation of tides into the general circulation model especially because tides have an important effect on the whole Mediterranean Sea (McDonagh et al. (2023)), including the Gibraltar Strait (Gonzalez et al. (2021)), which could in turn result in differences in the generation and characterization of internal variability. For instance, the recent works by Lin et al. (Lin et al. (2022), Lin et al. (2023c)) on the internal variability in the Bohai and Yellow Sea show that tidal forcing inhibits the generation of internal variability at large scales and that baroclinic instability might significantly drive the latter. Moreover, they suggest that the memory of the system is a critical factor in the generation of internal variability at large scales: the spectrum of the intrinsic variability with tides is less *red* than the one without. Since in a first-order autoregressive process, as the mathematical formulation of stochastic climate models, the *redness* of the spectrum is a measure of the memory of the system, i.e. a measure of how long an anomaly will last once formed, this comparison shows that tides, and winter conditions, tends to decrease the system's memory, inhibiting anomalies to persist and upscale. The study

of the importance of tides in the generation of internal variability would be interesting to be redone in a deep basin such as the Mediterranean Sea. Lastly, given the importance of submesoscales in several regions of the Mediterranean Sea (Trotta et al. (2017), CALYPSO (CAL), Solodoch et al. (2023)) future work could include the addition of submesoscale variability as a source of ocean internal variability.

*Data availability.* Results from these simulations are freely available at https://zenodo.org/records/10371026

*Author contributions.* All authors contributed to the design of this work. HvS, GL and NP formulated the initial problem idea, RB carried out all simulations and analyzed the output, and under the guidance of GL, RB developed the analysis tools. RAl authors contributed to the interpretation of the results

*Competing interests.* None

*Acknowledgements.* RB thanks GL, NP, and HvS for their guidance and mentorship throughout this work. Support for NP and GL work came from the H2020 EuroSea Project. The research presented in this paper was carried out on the supercomputer facilities of CMCC in Lecce (IT).

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
