# Peer review of "Internal and forced ocean variability in the Mediterranean Sea"

_EGUsphere, 2023_

## Author Comment (AC1)

**Comment on egusphere-2023-3076**
**Anonymous Referee #1**
*Referee comment on "Internal and forced ocean variability in the Mediterranean Sea" by Benincasa et al, EGUsphere, https://doi.org/10.5194/egusphere-2023-3076, 2024.*

We sincerely thank the referee for reviewing our manuscript and providing many constructive suggestions for improving the overall quality of the manuscript. A detailed report describing how the comments were addressed can be found below.
Reviewer's comment is shown in black color and italic font style. Our responses are shown in red color, and text from the manuscript, added or modified, can be identified by blue color and quotation marks.

- L4: I would suggest adding "different" to the realistic initial conditions to make it more clear. (as mentioned in L63)
  It has been added.

- L22: Is it possible to have riverine boundary impact for the intrinsic effects? Obviously atmospheric boundary is the most important, but if the riverine boundary could have an impact on the intrinsic effects. If there is large riverine input, it might lead to a different intrinsic effect.
  We thank the reviewer for their comment. Even though it is expected that differences in the riverine boundary conditions could have impacts on the intrinsic variability (i.e river plumes, river-induced coastal currents), we argue that at the Mediterranean basin scales the atmospheric forcing is the most effective driver of the internal variability at seasonal time scales. Alessandri et al. (2023) [4] have recently devised an ensemble approach for a coastal region in the Adriatic Sea influenced by riverine inputs. Their study indicates that uncertainties in river forcing do not exert a notable influence on the internal variability of sea level. In instances where such uncertainties do affect sea level, their impact is predominantly localized near the mouths of rivers.

- I believe L24-25: The sentence regarding the mesoscale eddies and flow instabilities is breaking the flow of the paragraph regarding Hasselmann's study. I would suggest moving this sentence to/towards the end of the paragraph. Maybe connect with energy cascade of meso to submesoscale eddies/flow.
  We propose the following modification from line 24 to ensure a smoother transition between paragraphs for better readability:

"The understanding of how internal variability can affect climate predictability, and justify the observed red profile of the climate variance spectra, was first achieved by Hasselmann (1976). To demonstrate the importance of internal variability in climate models, Hasselmann formulated a stochastic climate model whose main assumption is that the climate system may be divided into rapidly varying, random components and a slowly responding part. Climate variability is then shown to be due to the internal random components. The slow component behaves as an integrator of these inputs, whereas the fast component supplies the slow component with energy allowing the existence of internal variability in the climate system. Moreover, Hasselmann proved that climate variability would grow indefinitely without a stabilizing internal feedback mechanism. Consequently, the investigation of climate variability must be shifted from looking for positive to negative feedbacks that allow the climate system to reach stationarity in the absence of any external forcing. In the same years as Hasselmann's study, mesoscale eddies and flow instabilities were mapped for the first time in the ocean (Harrison and Robinson (1978), McWilliams (1996)) and the presence of intense ocean internal variability was verified to exist."

We have considered the reviewer's comment on the energy cascade from mesoscale to submesoscale. However, after discussion among the authors, we would prefer not to incorporate it into the manuscript. While we acknowledge the importance of energy cascades in ocean dynamics, we believe that, in this case, it could be misleading. We did not resolve the sub-mesoscales and our resolution is only partially effective at the mesoscales. We believe that discussions on energy cascade from and to smaller scales would be beyond the scope of our work.

- L49-50: What is the basis of Tang's study in the scale/grid resolution? It would be helpful to mention that the capability of high-resolution ocean models to resolve the subgrid scale processes compared to the coarse resolution models.
  We noticed that the sentence from L47 to L51 was a little convoluted. We suggest the following modifications to explicitly highlight the ability of high-resolution models to resolve subgrid processes, thus introducing more intrinsic variability:
  "Lastly, as regards the scale dependency of the internal variability, it was demonstrated by Tang et al. (2019) that additional intrinsic variability is produced by increasing the horizontal spatial resolution of ocean models from 1° to 0.04°. Furthermore, Tang et al. (2020) analyzed the ratio of the externally forced response and the internally generated variability in the South China Sea and showed that the external forcing is dominant at large scales, while most of the variability is internally generated at smaller scales."

- L80 mentions 0.1 degrees resolution for ECMWF. Coppini et al. (2023) mentions the same resolutions 0.125 degrees before 2020, and 0.1 degrees after 2020. Given the simulation period of Jan 2016 to Oct 2020, is there a possibility of a mismatch in the resolution?

  The ECMWF atmospheric boundary condition indeed experienced a change in the horizontal resolution in 2020 from 0.125° to 0.1°. However, we argue that our conclusions are not dependent on this change since all ensemble members were forced by the same atmospheric input. However, to demonstrate this point we repeated the same analysis for the year 2020 (see Supplementary material Figures S12 - S15). Even though we used a smaller ensemble of 16 members, the results were consistent and comparable, and the conclusions of our study still hold.

  We propose the following modifications in the manuscript at line 74 and following:

  "The model is forced by momentum, water and heat fluxes computed through bulk formulae using the operational analysis and forecast fields from the European Centre for Medium-Range Weather Forecasts (ECMWF). The ECMWF atmospheric boundary conditions have a horizontal resolution of 1/8° up to December 2020 and of 1/10° after. This change in the forcing's horizontal resolution is irrelevant since during the analyzed period all simulations are forced by the same atmospheric fields."

- L85-90: It is not very clear that the initial condition is used from the previous simulation (Assuming to my understanding). If this is the case, wouldn't the model carry some of the intrinsic variability through initial conditions to the next simulation? I believe this should be clarified.

  Each simulation is initialized using an independent analysis from the operational system (Clementi et al. 2019 [1]). The initial conditions are as realistic as they can be since analyses are the optimal combination of the numerical model solution and observations. We propose the following changes in the paragraph from line L83 to L89 to make the explanation of the strategy used in our study clearer:

  "Each simulation is initialized every three months starting from January 2016 to October 2020. The initial conditions are taken from the Copernicus Marine Service analyses (Clementi et al., 2019 [1]) and all simulations last up to December 2021, as explained in Fig. 2. The ensemble spread, related to internal variability, is generated by the different initial conditions."

- L164: I think one of the most important (albeit expected) results of this study is this line. I would discuss or emphasize this result more.

We sincerely thank the reviewer for their interest in our results. We believe that the large values of the ensemble spread at the thermocline level in summer are due to the strong vertical temperature gradient that amplifies small differences among the temperature fields of the simulations (explained in lines L157-L160). Moreover, differences in the velocity fields, such as the position or strength of eddies, can cause local upwelling or downwelling of the water column thus determining variations of the Mixed Layer Depth (MLD), and of the seasonal thermocline mid-depth. In the Mediterranean Sea, the thermocline disappears during winter and thus the relationship does not hold.
We suggest adding Fig. A1 and Fig. A2 below to Section S3 of the Supplementary material. Figure A1 shows the mean Mixed Layer Depth resulting from our ensemble in both summer and winter and it shows how dramatic the difference is between the two seasons. Figure A2 instead is a zoom-in on Figure 6 in the first 70 m where both the ensemble mean of the MLD and the MLD resulting from each simulation are provided to show the differences among them. These Figures should be called S10 and S11 in the Supplementary material given the present numeration.
We then suggest the following modification in line 164:

"We argue that, during summer, thermocline processes exhibit significant internal variability and the spread observed at the peak of the vertical temperature gradient may arise from various mechanisms. First, baroclinic instability localized there can generate internal variability. Secondly, changes in the position and strength of eddies can cause upwelling or downwelling, thereby influencing the mixed layer depth and consequently, the mid-depth of the thermocline (Figures S10 and S11 in the Supplementary material)."

- L168-170: should be rewritten to make it more clear. There seems to be a missing word or two.
  We propose the following modifications to lines 168-170:
  "The N/S for T is smaller than 1 up to 100 m (about 0.3 at the surface) and it increases with depth. In the surface layers, it shows greater values in winter, whereas at greater depths it attains systematically larger values (approximately equal to 6) in summer."

- I would argue in some part of the manuscript the number of ensemble simulations. Tang uses 4 simulation ensemble and Penduff uses 50 as large ensemble. It would be helpful to argue how the number 20 came up for the ensemble simulations? [See next point]
- Depending in this how many ensemble simulations would make a difference to be able to identify the intrinsic variability?

In ensemble studies, there is no absolute criterion to find the perfect ensemble dimension and, obviously, the more the ensemble members the better the estimation. We had a similar concern about the dependency of the accuracy of our estimation of the intrinsic variability on the number N of ensemble members, but we did not perform a rigorous study to define the most convenient N. However, initially, we had only 5 runs starting on January 1st of each year from 2016 to 2020, since we were following more closely the example by Tang et al. (2020) [2]. Then, to increase the accuracy of our estimation we added more simulations increasing N up to 20 in the way presented in Section 2 of the manuscript and we found no significant differences in the results, especially as regards the pattern of the intrinsic variability (please refer to Figures A3 and A4 at the end of this document).
We suggest adding Fig. A3 and Fig. A4 below to Section S1 of the Supplementary material. These Figures should be called S4 and S5 in the Supplementary material given the present numeration.
We suggest adding the following sentence at the end of the paragraph at line 91 to further clarify this point:
"It is important to notice that the choice of having an ensemble of 20 members was somewhat arbitrary, even if it was the largest number of members compatible with our computational resources and returned results similar to a smaller ensemble of 5 members (Figure S4 and Figure S5 in the Supplementary Material). "

- In general it is an important first step analysis towards understanding the intrinsic variability in the Mediterranean Sea. In addition, the tides would definitely add an interesting approach to the study and the results as mentioned in L215-. Overall all it is a good manuscript and I would recommend it for publication after the minor revisions mentioned.
  We sincerely thank the reviewer for their interest in our work and in the further developments that we proposed.

**References:**

[1] Clementi, E., Pistoia, J., Escudier, R., Delrosso, D., Drudi, M., Grandi, A., Lecci, R., Cretí, S., Ciliberti, S., Coppini, G., Masina, S., & Pinardi, N. (2019). *Mediterranean Sea Analysis and Forecast (CMEMS MED-Currents, EAS5 system)* (Version 1) [Data set]. Copernicus Monitoring Environment Marine Service (CMEMS). https://doi.org/10.25423/CMCC/MEDSEA_ANALYSIS_FORECAST_PHY_006_013_EAS5

[2] Tang, S., von Storch, H., and Chen, X.: Atmospherically forced regional ocean simulations of the South China Sea: scale dependency of the signal-to-noise ratio, Journal of Physical Oceanography, 50, 133–144, 2020

[3] Coppini, G., Clementi, E., Cossarini, G., Salon, S., Korres, G., Ravdas, M., Lecci, R., Pistoia, J., Goglio, A. C., Drudi, M., et al.: The Mediterranean forecasting system. Part I: evolution and performance, EGUsphere, pp. 1–50, 2023

[4] Alessandri, J., Pinardi, N., Federico, I., & Valentini, A. (2023). Storm Surge Ensemble Prediction System for Lagoons and Transitional Environments. *Weather and Forecasting*, *38*(9), 1791-1806.

**Added Figures:**

[Figure]

**Fig. A1:** Seasonal average of the Mixed Layer Depth in both winter **(a)** and summer **(b)** in the year **2021**. Please note the different scales used in the two sub-plots.

[Figure]

**Fig. A2:** Seasonally averaged vertical profile up to 70 m depth of the ensemble spread $\sigma_I$ (red) for potential temperature and of the vertical temperature gradient $-\nabla T_z$ (blue) in summer at Balea **(a)**, Ion **(b)** and EMed **(c)**. Horizontal lines indicate the seasonally

averaged Mixed Layer Depth: black corresponds to the ensemble mean, while orange indicates the ensemble members.

[Figure]

**Fig. A3:** Seasonal average of the ensemble spread with **N=5** for potential temperature T at different depth levels for the year 2021: at the surface **(a)**, at 30 m depth **(c)** and at 100 m depth **(e)** for winter and similarly in **(b)**, **(d)** and **(f)** for summer. Please note the different units used at different depths.

.

[Figure]

**Fig. A4:** Seasonal average of the ensemble spread with **N=5** for current speed v at different depth levels for the year 2021: at the surface **(a)**, at 30 m depth **(c)** and at 100 m depth **(e)** for winter and similarly in **(b)**, **(d)** and **(f)** for summer. Please note the different units used at different depths.

---

## Author Comment (AC2)

**Comment on egusphere-2023-3076**
**Anonymous Referee #2**
*Referee comment on "Internal and forced ocean variability in the Mediterranean Sea" by Benincasa et al, EGUsphere, https://doi.org/10.5194/egusphere-2023-3076, 2024.*

We sincerely thank the referee for reviewing our manuscript and providing many constructive suggestions for improving the overall quality of the manuscript. A detailed report describing how the comments were addressed can be found below. Reviewer's comment is shown in black color and italic font style. Our responses are shown in red color, and text from the manuscript, added or modified, can be identified by blue color and quotation marks.

*"In the paper by Benincasa et al. 20 simulations of the operational forecasting system of the Mediterranean Sea are used, through an ensemble approach, to assess the internal/intrinsic ocean variability. It is shown that such a variability is associated with the mesoscale activity and that, with the exception of the Adriatic Sea and the Gulf of Gabes, is larger than the response to surface forcing in all the Mediterranean Sea. Internal variability has a clear season cycle for temperature in the surface layers while, for marine current velocities, it is always dominant and largest at the surface.*

*The paper is clearly written and well organized and I have only a major concern related to the model resolution and the ability of resolving mesoscale instabilities. The model horizontal resolution is about 4km (1/24 deg) which may be not enough for a full development of mesoscale instabilities: in many Mediterranean areas the Rossby radius (Rd) is less than 8km (i.e. less than 2 model grid cells, see Fig.1 of Beuvier et al. 2012). This seems to be confirmed also by the Rd estimates provided in Grilli and Pinardi (1998) that in some cases are even smaller and closer to the model grid resolution. One may argue that the significant presence of internal variability found in this paper in some areas (e.g. the southern parts of the basin, see Fig.5) is just imputable to the model ability to fully resolve there mesoscales features. Indeed the southern parts of the basins are characterized by larger Rd values (see always Fig.1 of Beuvier et al. 2012). I suggest that the authors discuss such an important limitation of their methodological setup and the sensitivity of their results to horizontal resolution. As the authors themselves report at L48-49, the higher is the horizontal resolution the larger is the intrinsic variability."*

We sincerely thank the reviewer for pointing out this missing information. The reviewer's main concern of whether the model's horizontal resolution is sufficient to resolve mesoscales homogenously over the basin is very appropriate. Indeed, the model's

horizontal resolution of 1/24° allows it to be mesoscale eddy-permitting for most of the areas. As Beuvier et al. (2012) [1] detail, the Rossby radius of deformation in the Mediterranean Sea varies from 3 to 13 km. It is clear that an increase in the horizontal resolution (presently at 3.5 - 4 km) would imply higher values of internal variability. Nonetheless, we argue that within the model resolution limitations, the results are indicative of the relative importance of internal and forced variability.

We suggest the following addition to the *Model setup and simulations* after line 100:

"It is important to notice that the model we used is not everywhere eddy-resolving but mainly eddy-permitting. This is a consequence of the fact that the first Rossby radius of deformation in the Mediterranean Sea varies from 3 to 13 km (Beuvier et al., 2012) with larger values in the basin's interior and the southern areas. In contrast, in the Adriatic Sea and the Gulf of Gabes, the Rossby radius is generally smaller than the model's horizontal resolution thus possibly artificially decreasing the importance of internal variability."

And to the *Discussion and conclusions* after line 214:

"A limitation of our study is the underestimation of the internal variability stemming from the model's horizontal resolution of 1/24° which results in being too coarse for resolving mesoscale eddies everywhere in the Mediterranean Sea. Such underestimation could be particularly significant in the Adriatic Sea and the Gulf of Gabes where the spatial scales of mesoscale eddies tend to be smaller than the model's horizontal resolution. Thus, this could be an additional factor causing the small values of the ensemble spread in these regions. Nonetheless, the present study shows the importance of internal processes as opposed to the atmospheric influence compatible with the model resolution."

A process of revisions is suggested to address also the following minor concerns:

1. L9 "probably": it should not be so hard to assess whether or not the peak at 30m is really connected to the thermocline formation. Monin-Okubov depth?
   We thank the reviewer for pointing out this inaccurate statement. We have now eliminated the word from the abstract and we suggest to add at line 144 the following lines that reports on the 30 m thermocline depth findings in the literature:
   "In the literature, Hecht et al. (1988) [4] have described the Eastern Levantine thermocline seasonal variations showing it to be located between 20 and 40 m depth. Thus, we refer to 30 m as the average depth of the seasonal thermocline."

2. L42-43, just a curiosity: is there any quantification of the role of submesoscales in setting up intrinsic variability?
   Submesoscale variability has been mapped and studied at sub-basin scales, in limited areas, due to the resolution required (200 – 600 m). For instance, Trotta et al. 2017 [5]

studied the submesoscales associated with a large-scale anticyclonic gyre in the central Gulf of Taranto using realistic high-resolution submesoscale-permitting simulations obtained via multi-nesting techniques. CALYPSO (Coherent Lagrangian Pathways from Surface Ocean to Interior - https://calypsodri.whoi.edu/) was an international Research Initiative from 2018 to 2022 aiming at studying surface-to-interior 3D transport structures and pathways with advanced observing technologies in the Alboran Sea in the Western Mediterranean. Many of the projects and publications derived from it dealt with submesoscale variability and predictability, frontogenesis, and the link between mesoscales and submesoscales. Last but not least, Solodoch et al. 2023 [7] studied ocean variability in the Eastern Mediterranean Sea from basin-wide to submesoscales, with a particular focus on the latter.

We suggest adding the following phrase after line 225:

"Lastly, given the importance of submesoscales in several regions of the Mediterranean Sea (Trotta et al. 2017 [5], CALYPSO [6], Solodoch et al. 2023 [7]) future work could include the addition of submesoscale variability as a source of ocean internal variability."

3. L77, no tides: this is also reported in the conclusions at L215-216 and may represent an important future extension. But tides are important also in other areas apart Gibraltar (e.g. in the North Adriatic) where the intrinsic variability of this study may be underestimated.

We have now replaced the phrase in *Discussion and conclusions* at line 216 "within the Gibraltar Strait" with:

"[..] on the whole Mediterranean Sea (McDonagh et al., 2024 [8]), including the Gibraltar Strait (Gonzalez et al. 2021)."

4. L96-97: is there a specific reason to stop at 1000 m and not to perform the analysis for deeper layers?

The choice of 1000 m was justified by several factors. First, we indeed performed the analysis also for deeper levels of the water column, but we realized that the spread decreases with depth. Similarly, the influence of the atmospheric forcing decreases considerably at deeper layers, resulting in large values of the N/S that do not have the same meaning as the surface ones.

In conclusion, even if arbitrary, we believe that confining our analysis in the first 1000 m of the water column introduces a lower bound to the N/S for the largest area possible in the basin.

We added after line 98 a phrase to justify our choice:

"Limiting the analysis to the first 1000 m of the water column was justified in order to capture a meaningful N/S. The deeper levels showed decreased spread and decreasing atmospheric influence, as expected for the large-scale circulation. Despite its

arbitrariness, this boundary ensures a practical balance between vertical variability and statistical significance."

5. L108 \sigma_A formula: if \tau is the chosen period I am not sure I fully understand why t starts from 1 and there is a minus 1 in the denominator. My guess is that the total number of discrete timesteps making up the whole period should have been used instead.
In Equation 2 of the manuscript, the temporal standard deviation is computed for each grid cell over a season, i.e. 90 days, using Bessel's correction. However, given τ = 90, the difference between τ or τ -1 is insignificant. Thus, in line 108 we added:
"[..] is the temporal average of the ensemble mean over the chosen period τ, i.e. 90 days corresponding to a season."

6. L132: similar considerations on the \sqrt(\sigma^2) and RMSE formulae as the point above
Equation 4 is used to measure the dispersion of the ensemble, which is done to evaluate the quality of the spread of the ensemble of simulations (please refer to Fortin et al., 2014 [9] for a thorough explanation of why this is the right relation to consider). The equation links the square root of the average of the ensemble variance to the RMSE of the ensemble mean with respect to the reanalysis over the same period. Points below (above) the linear relation of Eq. 4 characterize an underdispersive (overdispersive) ensemble. It is important to notice that Eq 4 holds only for τ that goes to infinity and that is the reason we considered the entire 2021 for this estimation. Thus, in this case τ = 365 days, making the difference between using τ or τ-1 even more irrelevant than before.
We added in line 132:
"[..] since this relation holds for large values of τ and this the reason we considered τ = 365 days corresponding to the entire 2021."
We corrected also the typo in line 129: "members" must be changed with "mean".

7. L231: typo Eurosea project
We corrected the typo.

Lastly, we suggest adding [1], [4], [5], [6], [7] and [8] to the References of the manuscript.

**References:**

[1] Beuvier J., Béranger K., Lebeaupin Brossier C., Somot S., Sevault F., Drillet Y., Bourdallé-Badie R., Ferry N., Lyard F. (2012). Spreading of the Western Mediterranean Deep Water after winter 2005: Time scales and deep cyclone transport. Journal of Geophysical Research: Oceans, 117 (C7)

[2] Tang, S., von Storch, H., Chen, X., and Zhang, M.: "Noise" in climatologically driven ocean models with different grid resolution, Oceanologia, 61, 300–307, 2019.

[3] Tang, S., von Storch, H., and Chen, X.: Atmospherically forced regional ocean simulations of the South China Sea: scale dependency of the signal-to-noise ratio, Journal of Physical Oceanography, 50, 133–144, 2020.

[4] A. Hecht, N. Pinardi, A. R. Robinson, "Currents, Water Masses, Eddies and Jets in the Mediterranean Levantine Basin", Journal of Physical Oceanography, Vol. 18, No. 10, pp. 1320-1353, (1988), doi:10.1175/1520-0485(1988)018<1320:CWMEAJ>2.0.CO;2

[5] Trotta, F., Pinardi, N., Fenu, E., Grandi, A., & Lyubartsev, V. (2017). Multi-nest high-resolution model of submesoscale circulation features in the Gulf of Taranto. *Ocean Dynamics*, *67*, 1609-1625.

[6] Coherent Lagrangian Pathways from Surface Ocean to Interior - https://calypsodri.whoi.edu/

[7]Solodoch, A., Barkan, R., Verma, V., Gildor, H., Toledo, Y., Khain, P., & Levi, Y. (2023). Basin-Scale to Submesoscale Variability of the East Mediterranean Sea Upper Circulation. *Journal of Physical Oceanography*, *53*(9), 2137-2158.

[8] McDonagh, B., Clementi, E., and Pinardi, N.: The characteristics and effects of tides on the general circulation of the Mediterranean Sea, EGU General Assembly 2023, Vienna, Austria, 24–28 April 2023, 2023. https://doi.org/10.5194/egusphere-egu23-14117, 2023.

[9] Fortin, V., Abaza, M., Anctil, F., and Turcotte, R.: Why should ensemble spread match the RMSE of the ensemble mean?, Journal of Hydrometeorology, 15, 1708–1713, 2014.